# Tolerance of Iron-Deficient and -Toxic Soil Conditions in Rice

**DOI:** 10.3390/plants8020031

**Published:** 2019-01-28

**Authors:** Anumalla Mahender, B. P. Mallikarjuna Swamy, Annamalai Anandan, Jauhar Ali

**Affiliations:** 1Rice Breeding Platform, International Rice Research Institute (IRRI), Los Baños, Laguna 4031, Philippines; m.anumalla@irri.org (A.M.); m.swamy@irri.org (B.P.M.S.); 2ICAR-National Rice Research Institute, Cuttack, Odisha 753006, India; anandanau@yahoo.com

**Keywords:** rice, iron, soil, quantitative trait loci, genes, transporters, advanced genomic tools

## Abstract

Iron (Fe) deficiency and toxicity are the most widely prevalent soil-related micronutrient disorders in rice (*Oryza sativa* L.). Progress in rice cultivars with improved tolerance has been hampered by a poor understanding of Fe availability in the soil, the transportation mechanism, and associated genetic factors for the tolerance of Fe toxicity soil (FTS) or Fe deficiency soil (FDS) conditions. In the past, through conventional breeding approaches, rice varieties were developed especially suitable for low- and high-pH soils, which indirectly helped the varieties to tolerate FTS and FDS conditions. Rice-Fe interactions in the external environment of soil, internal homeostasis, and transportation have been studied extensively in the past few decades. However, the molecular and physiological mechanisms of Fe uptake and transport need to be characterized in response to the tolerance of morpho-physiological traits under Fe-toxic and -deficient soil conditions, and these traits need to be well integrated into breeding programs. A deeper understanding of the several factors that influence Fe absorption, uptake, and transport from soil to root and above-ground organs under FDS and FTS is needed to develop tolerant rice cultivars with improved grain yield. Therefore, the objective of this review paper is to congregate the different phenotypic screening methodologies for prospecting tolerant rice varieties and their responsible genetic traits, and Fe homeostasis related to all the known quantitative trait loci (QTLs), genes, and transporters, which could offer enormous information to rice breeders and biotechnologists to develop rice cultivars tolerant of Fe toxicity or deficiency. The mechanism of Fe regulation and transport from soil to grain needs to be understood in a systematic manner along with the cascade of metabolomics steps that are involved in the development of rice varieties tolerant of FTS and FDS. Therefore, the integration of breeding with advanced genome sequencing and omics technologies allows for the fine-tuning of tolerant genotypes on the basis of molecular genetics, and the further identification of novel genes and transporters that are related to Fe regulation from FTS and FDS conditions is incredibly important to achieve further success in this aspect.

## 1. Introduction

Rice is one of the most important staple foods and a significant source of daily caloric intake for more than half of the world’s population. It has been estimated that rice yields have to be increased by an additional 100 million tons to feed 9.1 billion people by 2050 [1]. However, the genetic gain for grain yield of rice has stagnated over the past decade, resulting in a plateau for rice production and productivity in favorable environments. At the same time, rice is facing multiple biotic and abiotic stresses in all rice ecosystems. Among the different abiotic stresses, such as drought, cold, heat, salinity, and acidic soils, soil nutrient deficiencies, and toxicities [2,3,4,5,6,7,8,9,10,11,12,13,14] cause significant grain yield losses. By understanding the genetic basis of several biotic and abiotic stress tolerances, significant progress has been observed in these areas and this helps in developing the stress tolerance of rice varieties. However, not much attention has been given to the development of rice varieties with tolerance of Fe deficiency soil (FDS) and Fe toxicity soil (FTS) [2,15,16]. 

Iron plays a vital role as a significant co-factor for several enzymes that are involved in mitochondrial respiration, photosynthesis, nucleic acids synthesis and repair, metal homeostasis, and in maintaining the structural and functional integrity of proteins and chlorophyll [17,18,19,20,21]. Therefore, understanding the impact of higher or lower availability of Fe on the healthy growth and development of rice plants is necessary. The soil is the primary source of Fe for plants and its optimum availability in the form of Fe^2+^ is essential for their healthy growth and development. Both a deficiency and an excess of Fe in the soil hinder several physiological functions in the rhizosphere [15,22,23,24]. About 30% and 18% of the global soil is Fe deficient and Fe toxic, respectively [25,26,27]. Altered soil redox potential, soil pH, soil fertility status, and intensity of Fe deficiency and Fe toxicity can cause significant grain yield (GY) reductions under Fe-deficient and Fe-toxic soil [28]. In severe FDS and FTS conditions, more than a 50% GY reduction is reported; however, complete crop failures have also been noticed during the early vegetative stage [28,29,30]. Rice plants usually maintain 60–300 ppm of Fe, while Fe-deficient plants may have 10–30 ppm of Fe. Under toxicity, Fe concentration may rise to 400–1000 ppm [27]. Among the reported tolerance mechanisms that are present in several rice cultivars (other than avoidance or exclusion through the root tip), the excess Fe is taken up in the vacuoles and apoplast, and followed by a physiological mechanism of detoxification of Fe-induced reactive oxygen species (ROS) by the activation of antioxidant enzymes [31,32].

Fe deficiency (FD) often occurs in alkaline soils. Fe becomes insoluble and immobile, leading to lesser root uptake from soil due to high soil pH; an excessive amount of calcium carbonate, nitrate, and heavy metals; poor aeration; unbalanced cation ratios; and, changes in temperature [6,7,19]. Further, a decrease in energy charges in the root cells of mitochondria for the secretion of mugineic acid (MA) leads to a depletion of Fe in shoots, and reduced mobility of the Fe transporting system inhibits chlorophyll formation [22,33,34]. Worldwide, FDS is a significant problem, and the majority of FDS exists in the southern and western United States, East and West Africa, and specific regions of Europe and Asia [35,36,37]. 

In contrast to FDS, FTS is common in acid sulfate soils and waterlogged conditions. Factors such as reduced cation exchange capacity, poor drainage, high sulfide content, and high organic matter content have led to excessive available forms of Fe in the soil [38,39]. A significant proportion of rice-growing area is affected by Fe toxicity (FT) in China, India, the Philippines, Thailand, Malaysia, and Indonesia, and several countries of West Africa [40,41,42]. Factors such as poor drainage, the release of organic solvents from plant roots into the soil solution, differences in landscape, reduction activity in oxidation-reduction potential, the formation of more hydrogen sulfides, enhancement of ionic strength, low soil fertility, low pH, soil organic matter content, interaction with microbial activities in soil and root, the relationship with other nutrients, and genetic variability of the genotypes influence the severity of FT in rice [43,44,45]. In low-pH paddy fields, the higher reduction activity of Fe^3+^ to Fe^2+^ increases Fe availability in the root zone with increased absorption and uptake of Fe content, thereby resulting in FT in rice [19,46]. Therefore, using proper soil and nutrient management practices, such as liming acid soils, improving soil fertility and soil drainage at certain plant growth stages, using manganese as an antagonistic element in the uptake of Fe^2+^, and deploying Fe^2+^-tolerant rice cultivars can help to avoid FT in rice.

Understanding the molecular and physiological bases of Fe regulation and transport of Fe from soil to plants is essential in developing rice varieties that are tolerant for FTS and FDS. However, little progress has been made in developing superior rice cultivars with this tolerance, mainly because of this complex trait that is governed by several component traits and is much affected by environmental factors. The development of rice cultivars with a tolerance of FDS and FTS would be helpful for resource-poor farmers in regions where such a deficiency and toxicity are rampant. In this review, we discuss recent progress in understanding the genetic and molecular basis of Fe absorption, uptake, and the transporting mechanisms in FDS and FTS, their interactions with other micronutrients, and tolerance traits that are associated with QTLs and genes through advances in breeding and genomic technologies for the improvement of FD- and FT-tolerant rice varieties. 

## 2. Chemistry of Fe in the Soil

Iron is the fourth most abundant element on Earth. It is found in the form of ferric oxides, hydroxides, and silicate minerals, which are not a readily available source of Fe to plants [16,22]. Fe content varies with the type and depth of the soil, ranging from 0.2% to 55% (20,000 to 550,000 mg kg^−1^), and the highest concentration is found at 2–15 cm [28,47,48]. For instance, the total Fe content in Indian soil ranges from 0.4% to 27.3% (40,000 to 273, 000 mg kg^−1^), but that accessible to the plant is extremely variable, from 0.36 to 174 mg, which depends on the soil, plant, and environmental factors [49]. Fe can be in either divalent (Fe^2+^) or trivalent (Fe^3+^) form in the soil. Fe^3+^ is not readily usable by rice plants as well as microbes due to the formation of insoluble oxides or hydroxides [50,51,52].

It is estimated that about one-third of Earth’s soil can be considered to be Fe deficient [53,54]. Factors such as soil aeration (aerobic condition/dry soil), organic matter content, pH, soil redox potential, microbial populations, etc., influence the solubility and availability of Fe to plants [38]. Among these factors, soil pH is the single most dominant one that affects Fe availability, and it is reported that a one-unit increase in soil pH above neutral pH can cause a 95% reduction in the availability of Fe [55]. Accordingly, saline, alkaline, sodic, and calcareous soils are naturally deficient in Fe, wherein Fe is converted to insoluble Fe-hydroxyl complex, thus limiting its absorption and uptake by roots [56,57]. However, in acid sulfate soils with pH less than 5, Fe is freely available in excess (2000 mg kg^−1^), causing FT to plants [58,59,60]. The amount of organic matter content in the soil and rate of its decomposition also influence Fe availability due to the excess production of bicarbonates and phosphates, which inhibits Fe absorption [27,61,62]. The increase in Fe concentration in the rhizosphere decreases the availability and absorption of other nutrient elements, such as phosphorus and potassium [63,64]. However, soil acidification by root exudates improves Fe availability and the efficient uptake of Fe^2+^ across the plasma membrane of root epidermal cells [65].

High root density and the presence of aerenchyma in the root zone regulate Fe^2+^ oxidation and the reduction process [38,66,67,68]. Soil microbial populations, such as *Pseudomonas*, *Bacillus megaterium*, *B. pumilus*, *Geobacter*, *Clostridium*, and *Bacillus* sp. also influence Fe availability by producing organic acids and regulating Fe oxidation and the reduction process in the root zone and mobilization of Fe-oxides. These facultative anaerobic chemo-organotrophic fungi possess various tolerance mechanisms, such as secretion of siderophores, Fe exclusion by the formation of an Fe plaque permeability barrier, enzymatic detoxification, modification of root architecture, efflux pumps, etc., which reduce the absorption and mobilization of Fe [6,69,70,71]. For example, after the depletion of oxygen in flooded soil conditions, Fe^3+^ acts as an electron acceptor for facultative anaerobic microorganisms, and the subsequent reduction of Fe-oxides and hydroxides can accumulate an enormous amount of Fe^2+^ in the soil solution [72].

## 3. Role of Fe in Plants

Iron plays an active role in the regulation of critical functions in plants, such as mitochondrial respiration, biosynthesis of nucleotides, photosynthesis, nitrogen assimilation, hormonal regulation, transport of nutrients, etc. [19,45,73]. Being a major cofactor for several enzymes, Fe helps in the regulation of protein stability and it also facilitates chemical reactions, such as hydration/dehydration, hydroxylation, redox-dependent catalysis, and photo-redox catalysis, which are involved in the detoxification of excess Fe; hence, it is one of the highly essential mineral elements for plant survival [34,74,75]. For instance, the excess concentration of Fe^2+^ within the plant cell may accelerate several redox reactions where Fe^2+^ acts as an electron donor and many antioxidant defense mechanisms of superoxide dismutase, catalase, polyphenol oxidase, and peroxidase play a significant role in protection from Fe-mediated oxidative damage. The activities of these antioxidants provides evidence to identify FT-tolerant rice cultivars [32,76]. Plants need approximately 10^−9^ to 10^−4^ M of Fe for optimal growth and development and to complete their life cycle [53]. More than 90% of the total Fe in plants is stored in the chloroplast to maintain the structural and functional integrity of the thylakoid membranes to actively carry out photosynthesis [19]. 

The frequent availability of Fe depends on two rice grown environments as aerobic (upland) and lowland rice eco-system. Generally, Fe in the soil is the form of oxihydroxide polymers, which has low solubility and limiting Fe for plant uptake and transport, mainly in calcareous soils. FDS is one of the major yield limiting factors and it causes interveinal chlorosis (IVC) of young leaves in the rice plant. The primary symptoms of Fe deficiency are IVC, which is affected by decreasing chlorophyll content, and these symptoms initially appear on the youngest leaves with yellow color and veins remaining in a green pattern, and subsequently decreasing shoot- and root-attributed traits and causing a significant decrease in yield and quality [33,56,60]. In severe cases of Fe deficiency, cell division occurs and leaves become white and lead to stunted growth [6,77]. 

In lowland conditions, the larger concentration of Fe^2+^ in soil condition leads to Fe toxicity Initially, FT appears as reddish spots on the tip of the lower leaves and it gradually spreads to the entire leaf blade; in addition, a black layer of Fe-S on roots has been found [29,78]. An excess amount of Fe^2+^ is initially involved in Fenton reactions and it generates hydroxyl radicals (OH^−^) and ROS, which causes irreversible damage to membrane lipids, proteins, and DNA, leading to significant oxidizing of the chlorophyll and subsequently decreasing photosynthesis [79,80]. Therefore, the absorbed excess amount of Fe that can induce cellular oxidative damage leads to alterations in morpho-physiological and yield-attributed traits [81].

Bronzing of leaves, root discoloration, and reduced root growth are traits visualized by excess Fe^2+^ [82]. The majority of visual symptoms of Fe toxicity are commonly observed at tillering and heading stage, but, in severe FT conditions, the effects can occur irrespective of any growth stage of the rice crop and may lead to complete crop failure [29,82]. Availability of excess Fe has antagonistic effects on other essential nutrient uptake that can lead to a significant yield reduction in rice [28,58]. For instance, Fe deficiency is identical with excess Mn and Fe toxicity with Mn deficiency; these two elements have a mutual antagonistic effect with the interaction of Fe in rice plants and antagonistic elements significantly reduce the uptake of nutrient elements P, K, Ca, and Mg with increasing Fe concentration [41,58]. Generally, in FTS, the availability and solubility of Fe exceed 300 mg kg^−1^, which has drastically disrupted the initial sensor site of root cell membrane structures; caused changes in the flowing of xylem and phloem; reduced enzyme activity, homeostasis, and metabolite synthesis; and, modified various genes and transporters that are involved in the uptake, transport, and translocation of Fe from soil to grain [17,34]. 

Leaf bronzing (LB) is one of the key symptoms appearing at the vegetative stage and it is associated with various morpho-physiological traits, such as shoot and root length, number of tillers and panicles, shoot and root dry weight, a modification in root architecture system (RAS)-attributed traits, days to flowering, grain yield reduction, chlorophyll content, enzyme activity, ethylene production, non-structural carbohydrates, photosynthesis activity, formation of Fe plaque, and aerenchyma that is majorly involved in Fe toxicity tolerance [2,13,58,62,83,84,85,86]. The high concentration of carbonates, phosphates, and nitrates in either hydroponics or the soil makes Fe unavailable to plants due to increased pH [52,87,88,89]. Bicarbonates reduce the uptake and mobility of Fe in the vascular tissue, stem, and petioles, which leads to Fe deficiency in the leaves [90,91]. Both deficiency and excess of Fe are detrimental to plants, but excess Fe is more damaging, as it causes irreversible damage since it triggers the production of large amounts of reactive oxygen species and free radicals, which affect membrane lipids and proteins and oxidize chlorophyll and subsequently decrease photosynthesis and grain yield [80,92,93]. Therefore, to avoid detrimental effects to the plants, different mitigation strategies are required to reduce the severity of the plants. 

## 4. Physiological Basis of Tolerance of FT and FD

Rice plants employ different approaches to overcome FD, such as reductase activity, secretion of root exudates, altering of root architecture, and activation of several metal transporters. FD tolerance in rice cultivars is strongly associated with increased secretion of root exudates, which helps to solubilize Fe and it makes Fe freely available for root uptake, which in turn is transported to different parts of the plant by various transporters [6,79,94,95,96,97,98]. Soil microbial activities, such as acidification, reduction, and secretion of Fe-chelating molecules influence the solubilization, absorption, and uptake of Fe from soil [99].

In FDS, Fe^3+^ is tightly bound to oxides or hydroxides at high pH, and it is not readily available for plant absorption [100]. The release of protons (H^+^) in the rhizosphere via H^+^-ATPase and phytosiderophore (PS) secretion in the root zone increases the availability of Fe. Increased protons lead to lower pH in the rhizosphere, thereby solubilizing Fe^3+^ to Fe^2+^, and this is transported via the plasma membrane by the specific transporting mechanism of membrane-bound ferric-chelate reductase enzyme and divalent cation transporters (IRT1) [22,52,101,102]. Recently, Krohling et al. [102] have suggested that, at pH 3.0, Fe is 1000 times more soluble than at pH 6.0. In the chelating strategy of phytosiderophores, they have a high affinity with Fe^3+^ without changing Fe^3+^ to Fe^2+^. The Fe^3+^-PS complex in xylem and phloem is transportation and translocation is performed by the root exudates and through the root cell membrane system with the help of various transmembrane transporters, such as *OsYSL1*, *OsYSL15*, *OsTOM1*, *OsIRT1*, *OsIRO2*, *OsIRT2*, and *OsNARMP1* and genes, such as *OsNAS1-3*, *OsNAAT1*, and *OsDMAS1* [6,21,45,52,96,103,104,105]. These molecular transporters and regulators demonstrate highly efficient Fe acquisition, uptake, and transportation from soil to above-ground plant organs under FD soil (Figure 1). Under FD conditions, most of the Fe is stored in the vacuole, and it will be made available for vital metabolic functions, such as photosynthesis and respiration mechanisms. Fe is transported from the vacuole to other parts by *OsPIC1*, *OsVIT1*, *OsVIT2*, *OsMIT1*, and *OsNRAMP* transporters during FD conditions [15,93,101,106,107]. However, for maintenance, Fe homeostasis is controlled by IDEF1 and IDEF2 during germination, *OsNAS2–3* and *OsNAAT1* after germination, and *OsFRDL1*, *OsDMAS1*, *OsYSL2*, *OsYSL15*, and *OsYSL18* are involved during vegetative and reproductive growth [6,15,44,45,103,108,109,110,111]. So far, in response to FD, the molecular signaling mechanism, energy conservation, and transporting of metals in mitochondria (MT) are not clear [112,113]. Two mitochondrial Fe transporters (MITs), FRO3 and FRO8, which are putatively involved in Fe^3+^ reduction in the MT membrane transported cytoplasmic Fe into MT in rice [92]. The knockout mutant analysis of MITs significantly exhibited reduced chlorophyll concentration, a higher amount of Fe in leaves, and altered respiration under FD [114]. Therefore, future research work on MT for the physiological and molecular regulation, transporting, and homeostasis of the Fe mechanism is required to understand FD tolerance in rice.

For adaptation to FT tolerance, rice plants use different tolerance mechanisms by avoiding excess amounts of Fe^2+^ through enzymatic oxidation at the root surface or releasing oxygen in roots, considered as a first defense mechanism, leading to the formation of root plaque as a physical barrier [32,83,84], internal distribution and storage of Fe^2+^ in shoots [16,60], and tolerance of excess amounts of Fe uptake involving free radicals and anti-oxidants via Fenton reactions [42,115,116]. However, excess Fe can also be retained in roots, which is another aspect of physiological root-based tolerance in rice [38]. Therefore, Fe exclusion or retention in roots played a major role in presenting or preventing oxidative stress, respectively.

Physiological and biochemical processes, such as oxidization of chlorophyll; respiration; releasing of root exudates; catalyzing hydroxyl radicals (OH^−^); ROS; antioxidant enzymes, such as ascorbate peroxidase (APX), dehydroascorbate reductase (DHAR), peroxidase (PD), superoxide dismutase (SOD), glutathione reductase (GTR), peroxidase (PD), and catalase (CA); and, anti-oxidants are involved in reducing oxidative stress in Fe toxicity [17,28,83,88]. For example, the radical scavengers of SOD and PD enzyme dramatically influence preventing Fe^2+^ uptake through the formation of hydrogen peroxide inside the leaves, which is a decidedly less active oxidizing agent than the radicals. The activities of detoxification enzymes APX, PD, CA, and DHAR are more highly expressed in FT-tolerant cultivars than in susceptible rice cultivars [117,118]. According to the availability of Fe in the soil, rice plants have developed a molecular mechanism through various transporters and genes to control absorption, uptake, transportation, and translocation. The key regulators of plasma membrane transporters *OsA1* to *OsA10*, *OsIRT1*, *OsIRT2*, *OsFRO2*, and *OsZIP1* to *OsZIP10* are directly involved in Fe capture [2,16,19,22,119]. In addition to that, fundamental helix-loop-helix transcription factors regulate the expression of FRO2 that is involved in regulating Fe uptake [45,120,121]. 

The major portion of Fe^2+^ (~90%) is typically immobile because of the mechanism of root apoplast and cation-exchange capacity in the cell wall, which are mainly affected by the alkalization process and alteration of physiological organic compounds, such as ethylene, ABA, auxin, and NO [122,123]. This stored form of macromolecule acts in favor of maintaining Fe homeostasis in rice [81,124]. The chief photosynthetic apparatus of the chloroplast is primarily affected by FD, which inhibits the electron transfer complexes (PSI, PSII, cytochrome b6f, and ferredoxins) and the biosynthesis of chlorophyll, whereas FT caused by an excess amount of Fe can generate oxidative damage in the chloroplast by releasing ROS via the Fenton reaction [60,75]. In FT conditions, tolerant plants play a key role in the formation of Fe plaque on the root, which acts as a barrier against Fe absorption [125,126,127]. This Fe plaque layer on the root surface, barriers at the root endodermis that are coated with hydrophobic suberin, thin leaf blade, and shorter root length, and inhibition of primary and lateral root growth are responsible for restricting excessive Fe absorption at the root level [6,82]. 

## 5. Phenotyping for FDS and FTS

Accurate phenotyping under FDS and FTS is essential to identify tolerant rice genotypes and to screen breeding materials [6,84,128]. Both field- and laboratory-based protocols and techniques with varying Fe concentrations and different compounds are being used to select FD- and FT-tolerant lines and also to select tolerant segregates in breeding programs (Table 1) [6,19,22,128,129,130].

IVC is one of the major disorders in FD, which leads to enormous losses in yield and grain quality [33,57,128]. Various traits such as rate of PS exudation into the rhizosphere, shoot and root biomass and number of adventitious roots, ferric reductase activity, photosynthetic rate, and root/shoot ratio have been used to identify FD-tolerant genotypes [7,59,65,128,131,132]. From the various phenotypic approaches, such as field-based, pot, and HNS (hydroponic nutrient solution) conditions, leaf chlorosis is a significant determinant of FD tolerance [15,24,59,133,134]. Of these methodologies, HNS has been used extensively to screen for FD tolerance in crops, such as soybean, chickpea, rice, wheat, maize, and okra [52,59,135,136,137,138,139]. 

Leaf bronzing is an essential trait for the identification of tolerant genotypes under FT conditions [28,42,83,140,141]. The first defensive mechanism against FT (altering root-attributed traits, such as increased root surface area, increased number of primary and lateral roots and root hairs, deep roots, increased root apex, high root density, and Fe-retaining in the root) played a significant role in the tolerance, and also helps in the uptake and loading of Fe into the xylem [38,79,142,143,144]. However, morpho-physiological and agronomic traits, such as shoot length, leaf number, coleoptile length, shoot dry weight, tiller number, root length, root dry weight, root number, grain yield and chlorophyll, shoot Fe concentration, root Fe concentration, proline, total phenol, total protein, and carbohydrate clearly differentiate the tolerance of susceptible rice varieties under FT [17,19,39,84,85,120,145]. In field evaluation of FT tolerance, 80 days after sowing (DAS) was considered as the best stage to record leaf bronzing score (LBS) for the amount of tolerance when compared with 40 and 60 DAS in two hotspot regions of FTS [30,40,84,146]. The higher genetic heritability and reliable correlation of LBS are significantly associated with GY reduction, suggesting that LBS is a practical approach for determining tolerant genotypes under FT [85], but, on the contrary, the results found a non-significant correlation between these two traits [140,147]. Therefore, the level of correlation mainly depends on factors, such as the intensity of Fe concentration, duration of Fe stress, type of soil environment, and type of genotype used in the experiment.

From the various phenotypic screening protocols, 300 mg L^−1^ of Fe concentration was found to be more suitable for screening FT-tolerant genotypes [39,84,87,145,148]. The LBS method has been used extensively in breeding programs for the development of FT-tolerant rice cultivars. Recently, color intensity variation has been incorporated into this method to make the measurement more quantitative [39]. The LBS scale has to be analyzed regarding the intensity of color variation of rice leaves and roots by using an RGB (red, green, and blue) image-based approach [39,149,150]. The RGB method measures the intensity of color changes in FT conditions by using an R (red)/G (green) index from the digital image of root and leaf [39,150]. The R/G index trait was found to be more useful for quantifying LBS, which is associated with FT and is quantitatively inherited with complex gene action.

Measuring Fe concentration in various plant parts is useful to understand the genotypic performance and mechanisms adopted by plants to tolerate FD and FT. Various advanced instrumental analysis and software technologies, such as inductively coupled plasma mass spectrometry (ICP-MS), atomic absorption spectroscopy (AAS), graphite furnace atomic absorption spectroscopy (GFAAS), synchrotron radiation micro X-ray fluorescence (SR-µXRF), and energy-dispersive X-ray fluorescence spectrometry (EDXRF) have been useful in measuring mineral elements [17,59,77,165,166,167,168,169,170]. Visual Minteq 3.1 and Pulse amplitude modulation (PAM) chlorophyll fluorescence [17,171] are useful for physiological and biochemical analysis of Fe concentration in leaf, shoot, and root; Fe content in nutrient solution; allocation of assimilates to various organs; structural characterization of elements from the leaf blade subapical region; images of chlorophyll fluorescence; accumulation of Fe content; and, quantification in various plant organs under FT. Using induction with the different sources of Fe, several researchers confirmed that FeSO_4_-EDTA is more toxic than FeCl_3_-EDTA and Fe-citrate, and this is highly useful in screening for FT [13,17,84]. However, Müller et al. [17] found that the highest translocation of Fe from roots to shoots was increased with Fe-citrate, whereas the lowest Fe translocation was noted with FeSO_4_-EDTA and FeCl_3_-EDTA. The increase in Fe concentration by Fe-citrate has not shown any visual symptoms of toxicity and Fe flux was found to be involved through Fe-citrate transporters [172]. In addition to that, stored Fe in the shoots was significantly associated with an increase in biomass production and was less toxic than other sources. Therefore, more Fe translocation and less Fe toxicity in the shoot indicate that shoot tissue plays a vital role in the FT tolerance mechanism [24,83,145]. 

## 6. The Genetic and Genomic Basis of FD and FT Tolerance

To exploit rice germplasm systematically, it is necessary to understand the genetic basis of FD tolerance and the molecular genetic information on associated traits. Here, we represented a schematic approach to the development of FD- and FT-tolerant genotypes (Figure 2). The different high-throughput phenotyping technologies and advanced genomics can enable us to dissect the many complex traits and they also lead to increasing yield productivity, especially in an integrated manner in breeding and genomics technology. For FD tolerance, to date, there has been a single report on control by a single dominant gene (*Ic*) in Prabhavati [173]; non-allelic complementary gene action among two sets of genes in ARC 10372, Cauvery, IET 7613, Prasanna, and Akashi [174]; and, purple coleoptile (*Pc*) was found to be closely linked with tolerance. Interestingly, to date, no QTLs have been identified for FD tolerance.

Genes such as *OsIRT1*, *OsIRT2*, *OsTOM1*, *OsNAS1*, *OsNAS2*, *OsNAS3*, *OsDMAS1*, *OsNRAMP1*, *OsNRAMP5*, *OsNAAT1*, *OsYSL2*, *OsYSL6*, *OsYSL13*, *OsYSL14*, *OsYSL15*, *OsVIT1*, *OsVIT2*, *OsSAM2*, and *OsYSL16* were reported to be significantly involved in enhancing PS secretion and the transport of Fe from roots to shoots and above-ground organs under FD [15,21,94,104,108,109,110,111,175,176,177,178]. Similarly, other transporters, such as DMA efflux transporter (TOM1), Fe^3+^-DMA transporter, root epidermis iron transporters, divalent metal transporters, Fe^3+^-deoxymugineic acid transporter, protocatechuic acid transporters, oligopeptide transporter, and Fe^2+^ transporters are responsible for Fe uptake, translocation of Fe, and the homeostasis mechanism in roots and shoots under FD [60,77,83,102,105,107,109,110,179,180]. The comprehensive list of FD tolerance genes and their respective functional roles appears in Figure 1 and Table 2. In addition, a few other elements, such as Fe-deficiency-responsive *cis*-acting elements (IDE) and binding factors *OsIDEF1*, *OsIDEF2*, *OsIRO2*, *bHLH*, *OsHRZ1*, *OsHRZ2*, and *OsRMC* have been associated with an increase in MA synthesis in the rhizosphere complex and this also maintains the homeostasis mechanism in rice plants under FD tolerance [6,74,109,178,181]. Ogo et al. [182] identified a total of 19 genes and transporters that are involved in long-distance transporting of Fe, metal absorption, Fe uptake, and translocation through tissue-specific expression profiling under FD conditions. Interestingly, *OsHAK22* is one potassium transporter gene that is associated with mugineic acid secretion for the tolerance of FD mechanism in rice [183]. By using an exogenous application of 24-epibrassinolide (EBR), *OsIRT1*, *OsYSL15*, *OsYSL2*, *OsNAS1*, and *OsNAS2* are significantly involved in Fe homeostasis, uptake, transport, and translocation mechanisms in wild-type rice cultivar Taichung 65 and d2-1 mutant rice [184]. The brassinosteroids (BRs) mediate the response to the FD tolerance mechanism by Fe deficiency-induced FRO (ferric reductase), Fe^2+^ transporters (IRT), and phytohormones (auxin, ethylene, nitric oxide, and mono oxide), which improves internal Fe availability and the translocation from roots to shoots [183,184,185]. 

In response to FT, the major genetic tolerance trait, such as LBS ranging from 0.05 to 0.88, has been significantly associated with FT and this has been used in developing FT-tolerant cultivars from the screening of different mapping populations, such as RILs, BILs, CSSLs, DHs, and F_2_ [31,38,39,62,117,154,160,186,187]. The heritability of FT is medium to high (43% to 85%), with an average of 73% in broad-sense heritability, and medium to low (2% to 45%), with an average of 31% under narrow-sense heritability, from the biparental mapping population and diverse rice germplasm. However, h^2^ns had a lower genetic variation than h^2^bs due to a lower proportion of additive variance, which could be explained by gene action in the inheritance traits [87,88,154]. By using biparental mapping populations, [152] noted that intensity of leaf color red (R)/green (G)index, root length, LBS, shoot length tiller number grain number per panicle and fresh shoot weight were associated with FT tolerance. The additive and dominance traits of SL and FSW and R/G index were non-significant through χ^2^ analyses, which indicate the presence of non-allelic gene interactions or epistasis. Among the various studies, African rice *Oryza glaberrima* cultivars have more tolerance of FT than Asian rice cultivars, and this can also be put into low-input systems [84,124,155,188,189,190]. Therefore, *O. glaberrima* could play a potential role in the development of breeding materials for the enhancement of FT tolerance in *O. sativa* cultivars.

Several QTLs for FT tolerance traits have been well characterized using different genetic resources of mapping populations, such as recombinant inbred lines (RILs) [31,83,159,186,191], backcross inbred lines (BILs) [83,155,186], doubled haploids (DHs) [117,192], chromosome segment substitution lines (CSSLs) [154], introgression lines of F_2_ and F_3_ [145,153,191,193], and wild rice accessions of *O. glaberrima* [155]. For easy understanding and future work, the QTLs reported for FT tolerance chromosome-wise are given in Figure 3. The work on FT QTL analysis is also found to be limited with 203 QTLs from 16 mapping populations. Those 203 QTLs were found to be distributed mainly on seven chromosomes (1, 2, 3, 4, 5, 7, and 11). The number of QTLs on different chromosomes varied from 1 to 39, and their PV ranged from 4.2% (*qCER*) to 47.2% (*qSDW*) [151,159]. In the diverse genetic background of populations, RILs, DHs, and F_2_ and F_3_ introgression lines were reported to have a high number of FT QTLs (>20), and they were mainly located on three chromosomes (1, 3, and 4). The QTLs on chromosomes 1, 3, 5, and 7 were found to be more consistent across the genetic backgrounds [155,191,194]. Out of these total QTLs, ten QTLs (one QTL for *qLBI*, two QTLs each for *qRDW*, *qSR*, and *qTN*, and three QTLs for *qNPQ*) on chromosome 1, one QTL (*qSIC*) on chromosome 2, four QTLs (two QTLs each for *qBFe* and *SFe*) on chromosome 3, and one QTL (*qCCI*) on chromosome 7 had more than 30% of the PV and this varied from 31.0% (qLBI_1) to 47.9% (*qSDW* and *qRDW*) [117,155].

The LBI trait is significantly expressed on chromosome 1 (RG345-RG381) and it has shown 32.3% of the PV [117,186]. In a similar way, Wu et al. [83] identified seven crucial QTLs for LBS (*qFETPX-1-1*, *qFETOX-1-2*, *qFETOX-2*, *qFETOX-4-1*, *qFETOX-4-2*, *qFETOX-7*, and *qFETOX-12*) on five different chromosomes (1, 2, 4, 7, and 12) and another three QTLs (*qFETOX-1-3*, *qFETOX-3*, and *qFETOX*-8) from chromosomes 1, 3, and 8 from two different genetic backgrounds of RILs of IR29/Pokkali and BILs of Nipponbare/Kasalath/Nipponbare, respectively. Matthus et al. [160] suggested that a significant threshold (−log10 P > 4.0) of single nucleotide polymorphism (SNP) markers on chromosomes 1 and 5 was strongly associated with leaf bronzing symptoms from the analysis of 329 diverse Asian rice accessions, which were collected from 77 countries using 44,100 SNP markers. The loci on chromosome 1 were co-localized with those of earlier reports of Dufey et al. [155] and Wu et al. [83]. Interestingly, the 172-kb LD block region on chromosome 5 located a candidate gene as putative vacuolar ATP synthase (*Os05g01560*), which is involved in Fe transport or subcellular Fe partitioning under FT tolerance in rice. Zhang et al. [129] identified 29 QTLs for FT and 31 QTLs for zinc toxicity (ZT) from the analysis of 222 *indica* rice accessions from 31 countries in Asia, Africa, and Latin America. Interestingly, ten QTLs were commonly detected on chromosomes 1, 2, 3, 5, 6, and 12 and two QTLs (*qSFe5* and *qSZn5*) were co-localized in the genomic region of 11.18–11.19 Mb on chromosome 5, associating Fe and Zn concentration in shoots under FT and ZT tolerance in rice. 

QTLs for LBS (*qFETOX-1-2* and *qFETOX*-1-3) on chromosome 1 were found to be co-localized with the QTLs reported by Dufey et al. [31,194] and Wu et al. [117]. The QTL (*qFETOX*-2) on chromosome 2 identified in an IR29/Pokkali population was close to an FT QTL reported by Wan et al. [193]. Therefore, the co-localized QTLs in the interval region of 36.8–41.0 Mb on chromosome 1 and 0 to 5 Mb on chromosome 3 confirmed FT tolerance [83,155]. These QTLs exhibited PV of 9.2–18.6%. Wu et al. [83] and Matthus et al. [160] also identified a common locus at the 26.7–29.4 Mb region that is responsible for FT tolerance in rice. 

YSL1, FRO, ZIP, NRAMP, and ferritin gene and transporter families are involved in the regulation of Fe uptake and homeostasis, intracellular targeting, and storage under FT conditions. Importantly, the physiological and metabolic-related functional genes *OsFER1*, *OsFER2*, *OsFPN1*, *OsFPN2*, and *OsNRAMP1* to *OsNRAMP8* are responsible for the maintenance of Fe homeostasis and storage of Fe in the vacuoles. Eleven genes (*OsNAS1*, *OsNAS2*, *OsYSL15*, *OsYSL16*, *OsNRAMP1*, *OsFRDL1*, *OsIRT1*, *OsYSL1*, *OsYSL15*, *OsVIT1*, and *OsVIT2*) are found to be involved in the transporting of Fe from long distance to leaves, stems, and grain and in the synthesis of PSs [18,45,46,110,120,172,199,201,207]. Similarly, *OsNRAMP1*, *OsNRAMP3*, *OsNRAMP6*, and *OsNRAMP5* are also involved in the accumulation and transport of other metal elements (Cd and Mn) in rice [107,120,197,206]. Therefore, it has been suggested that, by inhibiting the functions of these genes, an Fe-regulating mechanism under an excess amount of Fe might allow the activation of novel mechanisms under FT conditions [107,114].

The highly conserved transcription factor (TF) super family WRKY domain plays a major role in morphological changes in response to FT. Four TFs (*OsWRKY46*, *64*, *113*, and *55*) have been significantly involved in FT response in both root and shoot tissues [120]. The WRKY TFs may repress Fe translocation from roots to shoots under FT and also regulate root elongation. Further understanding of these WRKY TFs in response to the FT tolerance mechanism would make it possible to deploy the Fe regulatory networks that are controlled by these TFs. Several studies have confirmed that *OsDMAS1* and *OsVIT2* are significantly expressed in FT and FD treatments [20,107,114,208,209]. Bashir et al. [15] also noticed that DMA synthesis and the transporting gene *OsDMAS1* are significantly upregulated by FD and downregulated by FT in both roots and shoots. Further, through the microarray approach, these authors found that nine genes were upregulated by FT and downregulated by FD in roots. Among them, two bHLH transcription factors from the shoot and two small open reading frames (*sORFs*) were upregulated by FD and downregulated by FT [15].

## 7. Transcriptomics

The recent trend of whole-genome expression profiling by array-based technologies, RNA-sequencing, and differential gene expression (DGE) approaches plays a vital role in functional genomic analysis in the expression pattern of tens of thousands of genes, besides providing extensive knowledge of the diverse functional roles of transcripts of different plant parts under stress conditions [15,210,211,212,213]. Several efforts were made to predict physiological and molecular expression under different treatments of FD and FT by using a 110K microarray [15], 22K oligo-DNA microarray [214], and 44K oligo arrays [46] (Table 3). 

Based on the up- and down-regulation of genes, little information is available in response to FT. In contrast, FD is reported on the higher side with more up-regulation [15,96,199,216]. By using the new rice 110K array, Bashir et al. [15] identified 74 genes that were significantly upregulated by FD in both roots and shoots, whereas only 17 genes were downregulated by FT (500 μM Fe-EDTA) in both roots and shoots. Itai et al. [109] found that more than 20 genes have a similar expression pattern relative to MAs biosynthetic genes and methionine cycle-related genes *OsNAS1*, *OsDMAS1*, and *OsYSL15* at 3, 6, 9, 12, 24, and 36 h of FD treatments. Similarly, [182] identified 161, 739, and 446 genes that were upregulated by FD in the vascular bundle (VB), cortex (Cor), and exodermis plus epidermis (EP), respectively. The induced VB, Cor, and EP genes may facilitate the long-distance transport of Fe, radial transport of Fe through layers of the cortex, and Fe absorption from the rhizosphere. The Cor genes were involved in methionine synthesis in the Cor and they were actively involved in S-adenosyl-methionine, which is a precursor of NA and MAs in rice. With support from various researchers, DMA synthesis (*OsNAS1-2*, *OsDMAS1*, *OsTOM1*, and *OsYSL2*) and metallothionein genes (*OsIRT2*, *OsIDS1*, *OsIRO2*, *OsFRO2*, and *OsIDS1*) are up-regulated by FD and many of these genes regulate Cu and Cd toxicity and abiotic stress tolerance in rice [107,206,217]. Therefore, variations in the transcriptomics of FD and FT can also alter the availability of other metals and cause changes in the physiological and metabolic pathways in rice.

The recent emerging technology of microRNA (miRNAs) was explored in the control of Fe acquisition in roots under FT and FD conditions. Numerous miRNAs are involved in Fe homeostasis and the signaling of metal toxicity in response to deficiency and toxicity stress conditions [218,219,220,221,222]. Under FD and FT conditions, [162] identified the miRNAs miR156, miR162, miR167, miR168, miR171, miR172, and miR398 as being significantly involved in Fe homeostasis and they showed different expression profiling of Fe in root and shoot analysis. Among these, few miRNAs are significantly involved to regulate the metal transports from roots. For instance, miR156 has a relationship to other metal transports, like, P, N, S, and Mn [221], miR171 for N and Zn [221], and miR162 for tolerance to Cadmium. With the current trends in spliceomics studies in response to FT, Junior et al. [223] identified 123,682 and 127,781 different splicing junctions and the majority of these splicing events were noted in the intron. Retention was 44.1% in the control and 47.4% in Fe^2+^-stressed plants. Further, Junior et al. [223] analyzed post-translational modifications of proteins, for which they found that 25 genes are differentially expressed (five upregulated and 20 downregulated) in response to FT through phosphorylation (14.29%) and dephosphorylation (85.71%) mechanisms. Therefore, the understanding of alternative splicing events in the expression of post-translational modifications of proteins might provide great information for the coordination of molecular genetic networks and signaling cascades for tolerance of FT in rice. 

## 8. Proteomics 

The response to FD through proteomics studies in rice is minimal. Recently, Chen et al. [224] evaluated a Chinese rice variety (Yangdao 6) in FD conditions. A total of 73 proteins were identified as improved or reduced upon FD, and 63 of them were effectively reported under 2-DE and MALDI-TOF/MS. Among the 63 proteins, 40 were identified in rice leaves and 23 in roots. The identified proteins were strongly involved in different physiological pathways, such as photosynthesis, oxidative stress, ATP synthesis, energy metabolism, N metabolism, and cell growth or signal transduction in conditions of Fe deficiency. In corroboration with Xu and Shi [225], plant 14-3-3 protein (spot 115) plays a vital role in the activation of the plasma membrane H^+^-ATPase and it might be involved in iron mobilization. Therefore, the increase in fold expression of this protein was confirmed to be involved in response to FD. 

## 9. Epigenetics

Modification in histone and cytosine methylation by DNA methyltransferases is a key regulator, which is thought to cause gene silencing or gene activation at transcriptional and post-transcriptional levels [226,227,228,229,230]. In response to FD, one available report indicates that the epigenetic mechanism of DNA methylation pattern of methylation-sensitive amplified polymorphism (MSAP) technique have been followed in barley (*Hordeum vulgare* L) [231]. The significantly highest demethylation ratio was noted in comparison to the methylation ratio in Fe-deficient plants with respect to Fe-sufficient plants at 7, 9, 13, 15, and 19 DAS and the subsequent resupply of Fe-EDTA (100 μM) (RSF) at two and six days after (15R and 19R) treatments [231]. This indicates that FD induced more DNA demethylation than DNA methylation and this was also consistent with earlier reports that showed that abiotic stresses tend to demethylate genomic DNA [232,233,234]. Therefore, to understand the epigenetic mechanisms of histone modifications, DNA methylation pattern and chromatic structures may provide a better understanding of the molecular and physiological mechanism of imperative genes that are involved in Fe uptake and transportation, which may lead to adaptation under FD and FT conditions.

## 10. Agronomic Practices to Overcome FD 

In rice, Fe foliar sprays, a method of application at specific growth stages, combined with soil and water, and fertilizer timing crucially enhanced FD tolerance in rice. External Fe fertilizer application as foliar spraying of Fe-chelates, such as Fe-EDTA, Fe-sulfate, and Fe-salt; seed coating with Fe-EDDHA; and, soil applications have been used successfully in several studies to increase Fe availability and grain yield in FD conditions [34,100,235,236,237,238]. The effect of Fe fertilizer application in FD conditions is accompanied by an adaptation of the mechanism for uptake, transport, and translocation of Fe. However, this has shown inconsistent results due to an unavailable form of Fe, insufficient transportation of Fe concentration in shoots and roots, the mode of fertilizer application, and iron species. On the other hand, foliar application of Fe-chelates has some significant drawbacks: it requires several sprayings, the Fe-chelates can be washed off by rain, and they need a sufficient leaf area for absorption of Fe [239]. However, between the two methods of application, the application of foliar spray could be more efficient for correcting the Fe nutrient balance during the growth cycle and could aid in fast growth recovery from FD. Although soil fertilizers are cost-effective, they are not recommended, as they have the problem of immobilization from soil to roots and shoots. 

As compared to foliar applications of FeSO_4_ and EDTA·Na_2_Fe, EDTA·Na_2_Fe fertilizers can be effectively applied in the soil to reduce Cd accumulation in shoots, roots, and grains, which is a vital concern for human health. Moreover, these also increase other essential elements, such as Cu, Zn, and Mn, and grain quality traits in rice [240]. Soil EDTA·Na_2_Fe fertilizers could persistently provide adequate Fe^2+^ in the soil and subsequent transportation of Fe from shoot to grain through the transmembrane system, whereas FeSO_4_ can be easily oxygenated to Fe^3+^ and immobilized in the soil. However, several researchers have well documented and practiced foliar application and showed that Fe status in the shoot and shoot-to-root transporting of Fe nutrition played a vital role in Fe use in the root zone [241,242]. Particularly in anaerobic soils, Fe^2+^ has become more toxic at higher concentrations. In a previous study of Mandal and Halder [243], they applied a high amount of phosphorus nutrients, which significantly reduced the concentration of Fe^2+^ in the soil and also prevented the mobility of Fe in the embryonic shoots and roots [244,245]. Further, the application of potassium and zinc nutrient fertilizers reduces FT in the anaerobic soil, and it enhances yield [245,246]. Therefore, developing a genotype with tolerance and more efficiency in nutrient-extracting capacity would be more suitable for future breeding programs.

## 11. Phenotypic Screening and Breeding for FDS and FTS

The identification and development of FD- and FT-tolerant rice genotypes are considered to be affordable and practical approaches for improving yield and are considered as perpetual. Screening of traditional, improved, and mutant lines and biparental mapping populations at hotspot sites of soil with Fe toxicity and HNS with 10 to 2000 ppm of Fe^2+^ concentration and in the absence of Fe compounds with or without bicarbonates in nutrient solution [6,21,39,84,133,144,164] are the usual practices adapted by researchers around the globe. The methodologies that were adapted by several types of research worldwide to screen rice for FT and FD conditions are presented in Table 1. Few studies have been carried out to explore tolerant genotypes for FD conditions using HNS and hotspot sites. The outcome of the studies showed a reduction in plant growth parameters, such as plant height, shoot biomass, and root-attributed traits and physiological changes, such as less chlorophyll and photosynthetic rate, enzyme activity, and root Fe reduction in rice. From those studies, tolerant genotypes, such as Pusa-33, ARC 10372, IR36, and Cauvery, and moderately tolerant genotypes, such as WBPH 25, Prasanna, and Akashi were identified in comparison to control conditions under FD [98,174,247].

Under FTS, irrespective of the experimental conditions (pot/field), evaluating genotypes at 80 DAS is a more reliable and accurate stage for LBS than at 40 and 60 DAS in the two hotspot regions of FT soil at Africa Rice Centre [84]. This is in corroboration with earlier findings [30,40,146]. However, the phenotypic performance of tolerant and/or susceptible rice varieties may vary from one location to another and according to the type of soil. Sikirou et al. [84] noted that the susceptible rice genotypes IR64 and Bouake 189 showed a higher LBS with a GY reduction, whereas tolerant cultivars Suakoko 8 and WITA 4 showed a lower LBS and a GY reduction. The outcomes of some studies reported the susceptibility of Bao Thai [38,145,147] in contrast to the reports of Sikirou et al. [84] on FT tolerance. Differences in soil types might explain this type of contrasting reports. Creating FT conditions, supplementing normal soil with a higher amount of FeSO_4_, and collecting hotspot FT soil are cost-effective, but soil collection, washing, transporting, and filling of pots are more laborious. The major drawback of pot/controlled experiments using soil could be a major constraint in the transportation of FT hotspot soil from one country to another, which is restricted by quarantine issues. Using normal soil with a supplement of FeSO_4_ showed significant variations in morpho-physiological traits [62,158,194,248]. The authors found positive correlation with the essential traits of FT, LBS, shoot water content, shoot dry weight, root dry weight, relative variation in shoot and root dry weight, chlorophyll content index, and shoot iron uptake.

An alternative approach is the soil-free screening technique that is used by several researchers to screen different genetic resources, such as rice germplasm of diverse mapping populations by employing a varied concentration of FeSO_4_, duration of treatment, and stages of seedling growth for FT in rice. Morpho-physiological traits, such as LBS, grain yield, shoot water content, shoot dry weight, root dry weight, chlorophyll content index, blade Fe concentration, root-plaque Fe concentration, and shoot Fe concentration under different concentrations of Fe in HNS are crucial [39,155]. The PCA analysis of 24 selected rice genotypes, grain yield, and biomass are the parameters that differentiated tolerant genotypes from susceptible ones. Interestingly, genotypes Siam Saba, Cilamaya Muncul, Mahsuri, Margasari, and Pokkali had the lowest LBS, with high shoot Fe concentration and grain yield among all of the tested genotypes [39,152]. Therefore, the tolerant genotypes can store more Fe accumulation in the above-ground plant organs, and bring about more vigorous plant growth [62]. The identified FT-tolerant rice genotypes have been released in some iron-rich regions, but these same genotypes may not show tolerance in another instance of iron-toxic soil conditions due to the complexity of soil pH and toxicity and a deficient amount of other elements, such as Al, Mn, and Cd [249,250,251]. As compared with *O. sativa* accessions, *O. glaberrima* accessions showed more tolerance of FT in AfricaRice gene bank accessions [48,88,252]. It is noteworthy that *O. glaberrima* had an additional quality to adapt well under low-input and other stress-tolerant conditions [188,253]. Therefore, the significant variations that were observed among the tolerant genotypes for those morpho-physiological traits could be exploited in breeding programs for the identification and development of FT-tolerant genotypes.

Interestingly, FT-tolerant genotypes have exhibited their different amounts of tolerance in different environmental conditions. For instance, WITA 1 and Matkandu were moderately tolerant in FT hotspot soil in Korhogo, Côte d’Ivoire, but were sensitive at another location of FT soil in Guinea [78]. Similarly, Azucena, a *japonica* variety, has shown tolerance under 250 ppm Fe^2+^ for four weeks, whereas, at 1500 ppm Fe^2+^ concentration, it was reported as susceptible [192,254]. In addition to these, *indica* genotype Pokkali is sensitive in acid sulfate soil with chronic toxicity in the Philippines [255], but it has shown remarkable tolerance under 1500 ppm Fe^2+^ in hydroponics [254]. Therefore, this signifies the contradictory results of the responses of a genotype under different screening techniques, which reveals that, to understand the diversity pattern of rice under Fe toxicity, the intensity of Fe stress, duration, and adaptive strategies need more study. In summary, the screening of genotypes in the early stage in hydroponics could be superior. On the other hand, Sikirou et al. [84] suggested that pot screening techniques are more powerful for identifying donors and evaluating breeding materials, whereas hotspot regions of Fe-toxic soil would be more suitable for the screening of mapping populations and breeding lines for substantial results.

Improvement of grain Fe content is also one of the focus areas in rice breeding. However, recent evidence from [67] suggested that FT-tolerant aromatic rice variety Dom Sofid showed a significant increase in grain Fe concentration from 24% to 44% in acute (1500 mg L^−1^ Fe) and chronic (200 to 300 mg L^−1^ Fe) soil solution as compared with the control. Further, it has been suggested that, among the six varieties tested, Dom Safid from Iran had a clear response to FT tolerance with high grain Fe concentration.

## 12. Progress in the Development of FD- and FT-Tolerant Rice Varieties

The conditions of both FDS and FTS and associated traits regulating them are of a complex nature and they are strongly influenced by genotype × environment interaction and field heterogeneity, which make rice selection unsuccessful [34,83]. Increasing tolerance for the intake of an excess amount of Fe in rice plants and also enhancing Fe deficiency tolerance in Fe-deficient soil are important target areas for breeding programs. Progress in FT-tolerant lines has been sparked by various breeding strategies such as a pedigree breeding program, evaluation of various traditional landraces, and the selection of individual rice genotypes under toxic-iron soil and high concentration of FeSO_4_·7H_2_O in the nutrient solution [84,88,146].

The project Stress-Tolerant Rice for Africa and South Asia (STRASA) in collaboration with the International Rice Research Institute (IRRI) and Africa Rice Centre (AfricaRice) and their associated partners, such as national agricultural research and extension systems (NARES), the International Institute of Tropical Agriculture (IITA), and the International Center for Tropical Agriculture (CIAT) started breeding programs to develop FT-tolerant rice varieties. In collaboration, AfricaRice and its partners identified three highly tolerant rice cultivars (Mat Candu, Gissi 27, and Suakoko 8) and they were used as donors to develop FT-tolerant lines. However, those donors have significant drawbacks: tall growing nature, proneness to lodging, long duration, and moderate yield potential of 3 t ha^−1^ [256]. Drame et al. [88] evaluated a set of 172 rice genotypes that were collected from IRRI, CIAT, and NARES and they were grown in fields at four different sites (Valley du Kou and Banfora in Burkina Faso and Edozighi and Ibadan in Nigeria) and were evaluated in HNS in collaboration with IRRI and the University of Hohenheim. Based on the yield and FT score, 80 entries were identified and further tested in four different countries (Burkina Faso, Ghana, Guinea, and Nigeria) in participatory varietal selection (PVS) in rice gardens. As compared to Ibadan and Edozighi, the highest average grain yield was recorded by a farmer at Banfora sites (3071 kg ha^−1^). Four varieties, WAT 1046-B-43-2-2-2 (ARICA 8), IR75884-12-12-2-WAB1 (ARICA 6), ROK 25, and ROHYB 209-B-3-B-1, have significantly shown FT tolerance and they were selected at two experimental sites. However, ARICA 8 was selected from all four countries, whereas four cultivars, ARICA 6, SIK 9-164-5-1-3, WAT 1136-B-77-1-2-3, and WAT 1277-B-44-2-3-2, were selected in three countries [88]. These FT-tolerant rice varieties possess desirable traits, such as higher tiller capacity, medium plant height, early maturity, heavy panicle structure and shape, and higher grain yield. 

Based on conventional and molecular breeding approaches, the STRASA program has identified two FT-tolerant rice varieties, WAS 21-B-B-20-4-3-3 (ARICA 7) and WAT 1046-B-43-2-2-2 (ARICA 8), and were released them in the rainfed and irrigated lowland environments of Ghana, Burkina Faso, and Guinea. On the other hand, IR75887-1-3-WAB1 (ARICA 6) was released for rainfed lowland environments of Ghana. By exploiting the various screening methodologies, such as HNS and field and pot experiments and using partnership among several national programs and international research organizations, 18 FT-tolerant rice varieties were developed in Nigeria, Guinea, Burkina Faso, Liberia, Côte d’Ivoire, and Togo from 1977 to 2014 [13]. In addition to that, in improving breeding strategies, GIS tools are being used to map regions that are severely affected by Fe toxicity in several countries. Also, mutational breeding approaches have been exploited to identify and develop mutated lines. Pathirana et al. [144] identified FT-tolerant rice cultivars by culturing calli in a medium containing 9 μM of 2,4-dichlorophenoxyacetic acid (2,4-D), 0.5 μM of 6-benzylaminopurine (BA), and 0.1% concentration of methanesulfonate alkylating agent for 24 h. The most responsive variety, AT 353, was used as a donor in the FT tolerance breeding program and it was found to be a valuable resource for the screening and identification of tolerance for rice cultivars. For the identification of reliable donors, Mendoza et al. [158] screened diverse germplasm, including wild accessions, 24 improved, and traditional varieties of *O. sativa*, 18 from *O. glaberrima*, 10 from *O. rufipogon*, 13 from *O. sativa × O. rufipogon* derivatives, and 96 from *O. sativa × O. glaberrima* derivatives at Fe concentration of 400 ppm in HNS [156]. Among these, five promising varieties, BW267-3, Suakoko 8, IR9884, IR68544-29-2-1-3-1-2, and Azucena, and three *O. rufipogon* accessions (105909, 106412, and 106423) were found to have good tolerance and could be potential resources of valuable genes for FT tolerance in rice [156].

Progress in the development of FD- and FT-tolerant rice cultivars has shown moderately tolerant (MT) to tolerant (T) rice cultivars based on LBS, yield, and shoot Fe concentration. The FD- and FT-tolerant and moderately tolerant rice varieties are listed in Table 4 and Table 5. Through traditional pedigree breeding approaches, numerous national and international rice research breeding centers from Indonesia, India, Malaysia, the Philippines, and West and East Africa have developed FT-tolerant rice cultivars. Some of these identified FT-moderately tolerant rice varieties are still popular in lowland rice-producing systems. However, there is still a gap in improving yield potential with more FT tolerance for future breeding programs.

To date, few donors have been used to identify QTLs from biparental mapping populations, such as DHs, CSSLs, RILs, BILs, and F_2_ and F_3_ [31,46,83,145,151,154,155,186,192]. The identified FT tolerance QTLs and genes revealed the physiological and molecular tolerance mechanism concerning Fe absorption, uptake, distribution, and translocation or accumulation and provided useful information for breeding rice cultivars with tolerance of Fe toxicity. In the future, there is an urgent need for exerting research efforts to understand the genetics of FD tolerance and for developing an efficient screening protocol to strengthen future breeding programs for the development of FD tolerance. Therefore, the genetic variation in rice cultivars in response to FT and FD suggests that additional exploration of rice germplasm screening and genetic analysis are required for the improvement of specific genotypes for specific locations.

## 13. Conclusions

Iron (Fe) is a vital element for the maintenance of various cellular, metabolic, and physiological functions in all living organisms. The availability of Fe in the soil depends on factors, such as type of soil, pH, organic matter, and microbial activities that influence the significant role in the absorption, uptake, transport, and translocation of Fe from the soil to roots, shoots, and above-ground plant organs. The abundance of Fe in the soil varied significantly across diverse eco-geographical regions, from deficiency to toxicity for rice crop plants. Significant genetic variation exists for these conditions from the diverse genetic resources of rice germplasm upon their screening by adapting suitable methodologies that are relevant to deficiency or a higher accumulation of Fe. The emerging technologies facilitate science for a better understanding of the genetic, physiological, and molecular basis of Fe availability in the soil, and uptake, transport, and translocation to the various above-ground plant organs under FDS and FTS conditions. These advances aid in improving breeding strategies for enhancing tolerance in rice. However, understanding the physiological mechanism of Fe transport from the soil to the root tip zone, further increasing Fe content in the grain, and maintaining Fe homeostasis under FD and FT conditions remain substantial challenges for attaining the targeted tolerance with high grain Fe content in rice. A complete understanding of the genetic mechanism of Fe availability, uptake, and translocation under FDS is relatively lacking. As compared to FT, much more research progress has to be made for a better genetic understanding of FD tolerance, especially for developing superior breeding materials with higher grain yield. The correction of Fe content can be achieved by applying agronomic Fe fertilizers to FDS, and this will help in sustaining grain yield. Several significant and consistent QTLs were identified for FT tolerance from different genetic backgrounds of mapping populations along with selected donors that showed great potential for use in advanced breeding technologies. For a better understanding of the genetics of FD and FT tolerances, there is a need for a multi-disciplinary approach to identify key target tolerance traits with improved high-throughput screening techniques that can provide valuable insights into the underlying FD and FT tolerance mechanisms and Fe homeostasis in rice.

## Figures and Tables

**Figure 1 plants-08-00031-f001:**
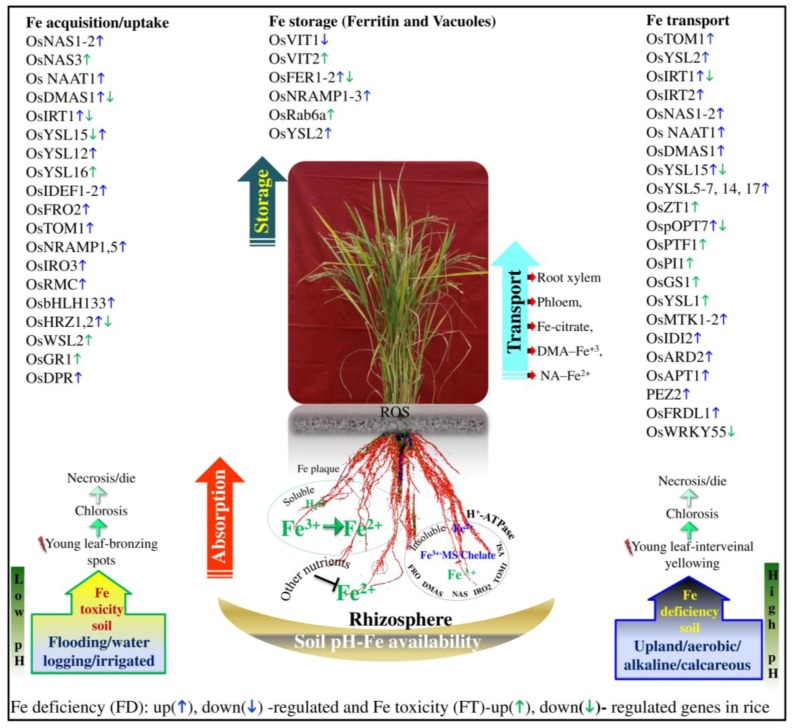
Diagram of Fe toxicity (FT) and Fe deficiency (FD) tolerance genes showing changes in transcriptional levels (up-regulation or down-regulation indicated by arrows) through microarray and transcriptomics studies.

**Figure 2 plants-08-00031-f002:**
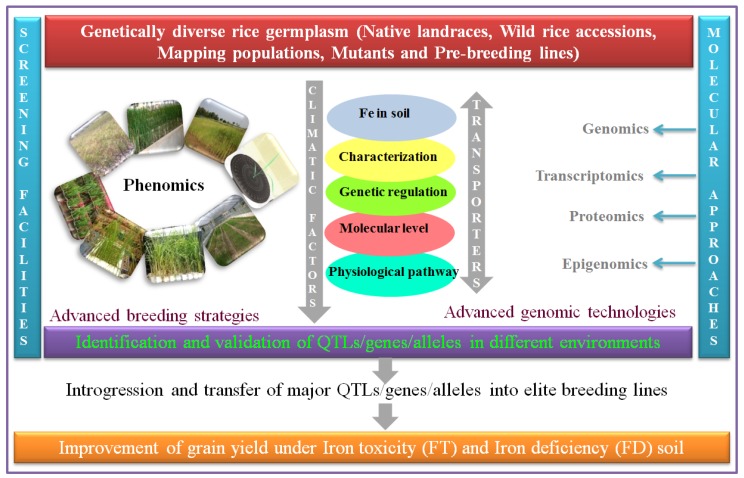
Schematic representation of innumerable phenotypic screening methodologies, and omics-based approaches to enhance FD/FT tolerance in rice cultivars.

**Figure 3 plants-08-00031-f003:**
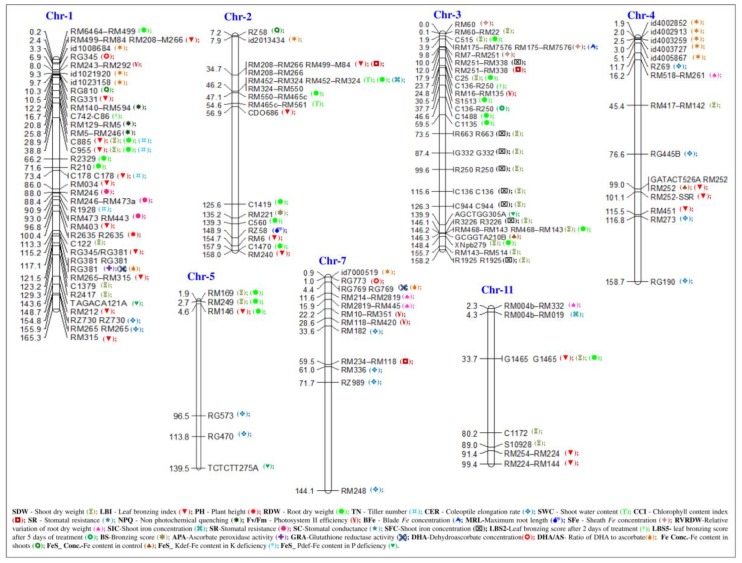
Comprehensive literature survey for the quantitative trait loci (QTLs) associated with Fe metabolism in rice.

**Table 1 plants-08-00031-t001:** Different methodologies used to screen germplasm against FD and FT in rice.

Method	Treatment	Concentration	Fe Compound	Genetic Resources		Adjusted pH	Days After Stress Imposed	References
				Genotypes	Populations			
HNS	FT	250 mg L^−1^	FeSO_4_	135	DHs	4.5	4 weeks	[117]
HNS	FT	250 mg L^−1^	FeSO_4_	96	BC_1_F_9_	4.5	1 week	[151]
HNS	FT	400 mg L^−^^1^	FeSO4	18	Inbred	5.5	2 weeks	[152]
HNS	FT	0.09 mM	Fe-EDTA	10	Inbred	5.0	1 week	[153]
HNS	FT	100 μM	FeCl_3_	8	Diverse accessions	5.6	2 weeks	[98]
HNS	FD	1 μM	FeCl_3_	8	Diverse accessions	4.6	2 weeks	[98]
HNS	FD	Without Fe	Fe-NaEDTA	4	Inbred	6.5	10 days	[7]
HNS	FD	0.1 mmol L^−1^	Fe-EDTA	2	Inbred	6.9	4 weeks	[128]
HNS	FT	250 mg L^−1^	FeSO_4_	164,	RILs	4.5	2 weeks	[31]
HNS	FT	5 mM	FeSO_4_	39	CSSLs	4.5	2 weeks	[154]
HNS	FT	400 mg L^−^^1^	FeSO_4_	2	Inbred	5.5	3 days	[85]
HNS	FT	250 mg L^−1^	FeSO_4_	220	DHs	4.5	3 weeks	[155]
HNS	FD	Without Fe	-	2	Inbred	5.1	3 days	[59]
HNS	FT	500 mg L^−1^	FeSO4	2	Inbred	5.1	5 days	[59]
HNS	FD	0.01 μM	Fe-EDTA	2	Inbred	5.9	2 weeks	[82]
HNS	FT	0 to 200 mM	FeSO_4_	51	Inbred	6.8	3 weeks	[26]
HNS	FT	400 ppm	FeSO_4_	161	Inbred	5.5	4 days	[156]
HNS	FT	0 to 640 mg L^−1^	FeSO_4_	6	Inbred	5.1	3 days	[157]
HNS	FT	7 μM	FeSO_4_	2	Inbred	4.0	1 week	[17]
HNS	FT	1000 ppm	FeSO_4_	2	Inbred	5.5	10 days	[67]
HNS	FT	0 to 3000 mg L^−1^	FeSO_4_	14	Inbred	5.0	4 weeks	[42]
HNS	FT	600 mg L^−1^ & 5 mg L^−1^	FeSO_4_ and Fe-EDTA	97	F_8_	5.0	2 weeks	[158]
HNS	FT	300 mg L^−1^	FeSO_4_	20	Diverse accessions	3.0–4.5	3 days	[145]
HNS	FT	1.79, 7.16, & 14.32 mM	FeSO_4_	244	RILs	5.0	12 hours	[159]
HNS	FT	1000 ppm	FeSO_4_	329	Diverse accessions	5.5	2 weeks	[160]
HNS	FT	300 mg L^−1^	FeSO_4_	211	Inbred	5.0	5 days	[161]
HNS	FD	Without Fe	-	2	Inbred	6.0	1 week	[162]
HNS	FT	25, 50, & 75 mg L^−^^1^	FeSO_4_	2	Inbred	5.5	3 weeks	[28]
HNS	FT	300 ppm	Fe-EDTA	4500	Mutants	3.0	10 days	[148]
HNS	FT	200 mg L^−1^	Fe-EDTA	4	Inbred	5.6	3 weeks	[76]
HNS	FT	125 mg L^−1^	FeSO_4_	1	Inbred	5.0	2 weeks	[46]
HNS	FT	400 mg L^−^^1^	FeSO_4_	23	Inbred	5.6	2 weeks	[39]
Field experiment	FT	HTS	-	2	Inbred	6.7	3 weeks	[28]
Field experiment	FT	2030 mg kg^−^^1^	-	2	Inbred	3.9	3 weeks	[85]
Field experiment	FT	40 to 140 mg L^−1^	FeSO_4_	2	NILs	5.0	3 weeks	[147]
Field experiment	FT	750 ppm	FeSO_4_	5	DHs	5.0	3 weeks	[163]
Field experiment	FD	7.2 2 mg kg^−1^	FeSO_4_	1	Inbred	8.2	6 weeks	[164]
Pot experiment	FD	14.0 mg kg^−1^	FeSO_4_	1	Inbred	8.2	5 weeks	[164]
Pot experiment	FT	HTS	FeSO_4_	5	Inbred	4.5	3 weeks	[84]
Pot experiment	FT	HTS	-	172	Inbred	5.0	3 weeks	[88]
Pot experiment	FT	HTS	-	40	Inbred	5.1	2 weeks	[88]

**Table 2 plants-08-00031-t002:** List of identified transporters, homeostasis, and translocation of *Fe*-regulated genes and their putative functions.

S. No.	Genes	Location	Function	References
1	*OsNAS3*	Shoot ^▲^	Involved in mugineic acid pathways (MAs) to transport from soil to root	[180]
2	*OsTOM1*	Root ^▲^	Mediates the efflux of DMA into rhizosphere and followed by formation of Fe^3+^-MA complexes	[96]
	*OSIRT1* and *2*	Root ^▲^	Integral membrane Fe^2+^ transporter	[45,195,196]
3	*OsIRO2*	Root and shoot	DMA biosynthesis	[196]
4	*OsNAS1-2*, *Os NAAT1*, and *OsDMAS1*	Root and shoot ^▲^	Fe acquisition and translocation	[45,180]
5	*OsNRAMP1*	Root and shoot ^▲^	Transporation of Fe from roots to aerial parts, including rice grains	[46,197]
6	*OsYSL2, 15, 16*	Root ^▲^	Phenolics efflux mechanisms, which can bind with Fe^3+^ for uptake and long-distance Fe transport	[45,46]
7	*OsZIP4*	Root ^▲^	Zinc-transporting protein involves Fe transport and homeostasis	[46]
8	*OsYSL2*	Root ^▲^	Fe-NA transporter and Fe accumulation in seeds and translocation of Fe into the grain	[108]
9	*OsFRO2*	Root ^▲^	Absorption and uptake of Fe from soil to roots	[24,198]
10	*OsFRDL1*	Shoot and flower ^▲^	Long-distance Fe transport and homeostasis	[108]
11	*OsPIC1*	Chloroplast ^▲^	Fe transport from root to chloroplast	[107]
12	*OsMIR*	Shoot ^▲^	Fe homeostasis	[199]
	*OsNRAMP5*	Shoot ^▲^	Fe translocation	[197]
13	*OsDMAS1*	Root and shoot ^◈^	Vacuolar Fe transporter from roots to shoots	[20,107,200]
14	*OsVIT1,2*	Leaves and seeds ^◈^	Vacuolar Fe transporter	[45,106,107]
15	*OsFER1*	Aleurone layer ^◈^	Vacuolar Fe transporter and homeostasis	[201]
16	*OsFER1* and *OsFER2*	Leaves ^◈^	Increased in leaves under excess ferrous iron conditions	[19,59,201,202]
17	*OsIRT1*, *OsNAS1-3*, *OsNAAT1*, *OsFRO2*, and *OsDMAS1*	Root and shoot ^ 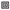 ^	Biosynthesis of chelates and PS, expressed in phloem companion cells to contribute to iron translocation and transport of Fe from roots to shoots	[24,45,114,203]
18	*OsFRDL1*	Root and shoot ^ 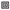 ^	Fe^3+^-citrate complex localized on pericycle cells of roots to contribute to iron translocation to shoots	[172]
19	*OsFRO2*	Root ^ 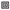 ^	Changing Fe oxidation state	[201,204]
20	*OsWRKY55*, *OsWRKY46*, *OsWRKY64*, and *OsWRKY113*	Root and shoot ^ 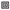 ^	Repress Fe translocation to shoots by regulating root elongation	[120]
21	*OsIDEF1*, *OsIDEF2*	Leaves and roots ^ 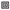 ^	Metal transporters and through the binding process of TF IDE-binding factor 1 and 2 genes	[74,104]
22	*OsYSL2*, *OsYSL15*, and *OsYSL18*	Root ^ 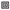 ^	Fe-chelate transporter, expressed in phloem companion cells, to contribute to iron translocation through the phloem	[45,203]
23	*OsNAS1*, *OsNAS2*, *OsNAAT1*, *OsDMAS1*, and *OsYSL15*	Root ^ 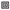 ^	Regulate PS-mediated Fe uptake and homeostasis	[181,196]
24	*OsYSL16*	Epidermis and cortex ^ 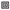 ^	Acquisition and inter-organ transport of Fe xylem-to-phloem	[94,205]
25	*OsYSL18*	Lamina joints, flower ^ 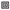 ^	Fe^3+^-DMA transporter involved in DMA-mediated Fe dispersal in reproductive organs	[205]
26	*OsYSL5-7*, *-14*, and *-17*	Epidermis, cortex, and stele ^ 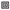 ^	Fe transport from root to shoot and grain	[45]
27	*OsYSL12*	Cortex and stele ^ 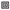 ^	Fe transport from root to shoot and grain	[45]
28	*OsNRAMP1*	Shoot ^ 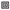 ^	Fe transport from root to shoot and highly expressed in Fe deficiency	[206]

^▲^ Fe transport; ^◈^ Fe storage; ^
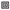
^ Fe uptake.

**Table 3 plants-08-00031-t003:** Up- and down-regulation of a list of genes for FD and FT conditions.

S. No.	Microarray	Shoot		Root		Reference
		Down-Regulation	Up-Regulation	Down-Regulation	Up-Regulation	
1	44K	-	-	325 ^a^	1068 ^a^	[109]
2	110K	318 ^a^ & 1655 ^b^	258 ^a^ & 2480 ^b^	2655 ^a^ & 116 ^b^	1509 ^a^ & 36 ^b^	[15]
3	44K	-	1346 ^a^	-	80 ^b^	[215]
4	44K	280 ^b^	519 ^b^	-	-	[46]
5	60K	195 ^b^	645 ^b^	2304 ^b^	1656 ^b^	[118]

FD ^a^, FT ^b^.

**Table 4 plants-08-00031-t004:** List of genotypes identified for FT- and FD-tolerant rice cultivars.

S. No.	Cultivar/Variety	Tolerance Level	References
1	Cauvery, ARC 10372	T ^a^	[174]
8	IPB Kapuas 7R, IPB Batola 6R, IPB1 R Dadahup, IPB Batola 5R, Indragiri, Margasari, and A. Tenggulang	T ^b^	[257]
10	Kapuas	T ^b^	[258]
11	Tuljapur	T ^a^	[174]
13	Inpara 2, B13144-1, Cilamaya Muncul, and Margasari	T ^b^	[152]
14	B13144-1-MR-2	T ^b^	[259]
15	Cilamaya, Siam Saba, Mahsuri, Pokkali, and Awan Kuning	T ^b^	[39]
16	TOX 85C-C1-15-WAS 1, TOX 85C-C1-16-WAS 1, WITA 3, TOX 3069-66-2-1-6, FKR 19, WITA 4, CK 4, CK 73, BW 348-1, TOX 4216-25-2-3-1-3, WAT 1059-B-51-2, WAT 1282-B-3-3, WAT 1131-B-26-2-1-2, Nerica-L19, ARICA 6, ARICA 7, and ARICA 8	T ^b^	[28,40,88,146,260]
17	IR61246-3B-15-2-2-3, IR61612-3B-16-2-2-1, IR61640-3B-14-3-3-2, WITA 7, Suakoko 8 (ROK 24), TCA 4, and Azucena	T ^b^	[145]
20	CK4 and Tox4004-8-1-2-3	T ^b^	[42]
21	OG 7206, TOG 6218-B, and TOG 7250-A	T ^b^	[84]
22	Ghanteswari, Mahanadi, Surendra, Bhanza, Lalat, Daya, Keshan, Rajeswari, Tejaswini, Sankar, Bhuban, Uphar, Khandagiri, Udayagiri, Manika, and Rudra	T ^b^	[26]
23	Suakoko 8	T ^b^	[186]
24	BR IRGA 414, IRGA 419, and BRS AGRISUL	T ^b^	[157]
25	PBN1 (Prabhavati)	T ^a^	[173]
26	ISA-40 and PSQ-4	T ^b^	[245]
27	Pusa-33	T ^a^	[98]
28	EPAGRI 108	T ^b^	[59]
29	IR36	T ^a^	[247]
30	Dom Sofid	T ^b^	[67]

Fe deficiency ^a^; Fe toxicity ^b^, T = tolerant.

**Table 5 plants-08-00031-t005:** List of genotypes identified for FT and FD moderately tolerant rice cultivars.

S. No.	Cultivar/Variety	Tolerance Level	References
1	Mahsurian	MT ^b^	[259]
2	WITA 1 and Matkandu	MT ^b^	[78]
3	Prasanna	MT ^a^	[174]
4	Akashi	MT ^a^	
5	WBPH 25	MT ^a^	
6	Inpara 3	MT ^b^	[152]
7	IET7613	MT ^a^	[145]
8	Phalguna	MT ^b^	[145]
9	PSBRc 18	MT ^b^	
10	WITA 1 and WITA 2	MT ^b^	
11	Mahsuri	MT ^b^	
12	CG14	MT ^b^	[38]
13	I Kong Pao and Sahel 108	MT ^b^	
14	ITA 306 and ITA 320	MT ^b^	
15	IR74 and Mahsuri	MT ^b^	[186]

Fe deficiency ^a^; Fe toxicity ^b^, MT = moderately tolerant.

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
