# Peer review of "Tolerance of Iron-Deficient and -Toxic Soil Conditions in Rice"

_plants, 2019, doi:10.3390/plants8020031_

Round 1
Reviewer 1 Report
A well executed review one the importance of iron deficiency and -tolerance in rice.
A table of abbreviations with their explanations right after the abstract would make the manuscript more clear and readable.
Author Response
Reviewer 1
Comments and Suggestions for Authors
A well-executed review on the importance of iron deficiency and -tolerance in rice.
A table of abbreviations with their explanations right after the abstract would make the manuscript more clear and readable.
Response: Thanks for your suggestions. We have now included all the abbreviations and expanded them in the revised manuscript.
Reviewer 2 Report
Graphical abstract, the current version contains way too much text, and is difficult to grasp the main points. Suggest reducing the level of details in the graph, such as listed examples of “other nutrients”, which could be removed from the graph. It would be nice to replace the current black and white rice plant photo with some color photos of the symptoms showing leaf bronzing and interveinal chlorosis. This will provide a more direct impression of the severity of FTS and FDS, respectively. The text “INTEGRATED APPROACH” is placed inside the photo, making it disconnected from the accompanying information to the right. Please relocate the text to make them better connected. Also, the orientation of some text is awkward, and please fix them in a way to make them easy to read.
Throughout the manuscript, “Fe+2” and “Fe+3” should be changed into “Fe2+” and “Fe3+”.
The Introduction section contains many contents that are redundant with the later sections in the manuscript. The introduction section should provide the necessary background knowledge on how significant FD and FT issues are on rice production, as well as a logical outline of contents of the manuscript. For example, many detailed description of FD and FT symptoms in the “1. Introduction” section should be moved and incorporated into “3. Role of Fe in plants”
The manuscript can be greatly condensed by reducing the aforementioned repetitiveness, and removing many uninformative statements, which include 1) writing down long lists of factors or gene names may be involved in responses to FT or FD without actually providing in depth discussion on the roles they play; 2) listing trivial numbers from other people’s study without giving a comprehensive summarization; 3) very general statements that is not directly related to FT and FD study, such as on line 540 to 542 “Among them, few proteins are validated for their biological functions, and many of them were unrevealed, and those may not be regulated at post-translation levels”.
The manuscript needs to highlight the take-home messages the authors want to deliver to their readers. The current version clearly lacks that, and may leave the audience feel confused or lost after reading through it.
Line 46, suggest change “adding” into “additional”.
Line 63 to 65, the sentence “Affected by soil pH, reduced mobility of nutrients within the soil, and reduced soil-microbial interactions, overall poor soil health conditions affect the growth and survival of rice plants” is too general and not very informative. It could be removed.
Line 65, suggest change into “About 30% and 18% of the global soil is Fe deficient and Fe toxic, respectively”
Line 68, what is GY? I guess it means grain yield, but it is not defined in the entire manuscript.
Line 88 to 93, it is confusing that the authors suddenly shifted the topic from FD to FT, while the prior portion of the paragraph has dedicated to FD. Why not move the FT contents to the following paragraph, where FT is discussed in greater details. Also, the sentence “Excess soluble Fe causes reduced cation exchange capacity, poor drainage, high sulfide content, and high organic matter content, leading to increased Fe availability, absorption, and uptake from the soil” is not interpretable. It is unclear from the writing whether high Fe content is the cause or consequence.
Line 101, please elaborate on “antagonistic effects on other essential nutrient uptake”.
Line 130, missing “the” in front of “Earth”. On the same line, please change “Fe oxides” into “ferric oxides”.
Line 132, I cannot understand “0.2% to 55% mg kg-1”. Does this mean 0.002 to 0.55 mg per kg? Same issue applies to the following sentence. Please also be consistent with unit for iron content in soil. On Line 146, the unit used becomes mg per L.
Line 149 to 150, the sentence “The higher concentration of Fe in the soil also decreases the availability and absorption of other nutrient elements such as phosphorus and potassium” seems mis-placed. It would read more fluent if following the sentence ending in “causing FT to plants” on line 147.
Line 169 to 172, the sentence “Being a major cofactor for several enzymes, Fe helps in the regulation of protein stability and also facilitates chemical reactions such as hydration/dehydration, hydroxylation, redox-dependent catalysis, and photo-redox catalysis, which are involved in the detoxification of excess Fe” implies Fe facilitates chemical reactions that detoxify excessive Fe, which does not sound intuitive. Providing an example of such reaction would help readers to understand the process. In addition, this sentence is highly redundant with the sentence on line 176 to 177, which says “Fe acts as a cofactor for several enzymes, helps regulate protein stability, and also facilitates the several chemical reactions that are involved in detoxification”. Please delete either one.
The sentence from line 177 to 182 “Both deficiency and excess of Fe are detrimental to plants, but excess Fe is more damaging as it causes irreversible damage since it triggers the production of large amounts of reactive oxygen species and free radicals, which affect membrane lipids and proteins and oxidize chlorophyll and subsequently decrease photosynthesis and grain yield [80–82]. Because excess Fe is detrimental to plants, its mitigation through various strategies is necessary.” Should be moved to the end of this section, following the description of FD and FT symptoms and mechanisms.
Section “4. Physiological basis of tolerance of FT and FD” How are FT and FD defined in rice cultivars?
Line 215 to 216, “Increased protons lead to lower pH in the rhizosphere, thereby solubilizing Fe+3 to Fe+2” implies by lowering pH alone, Fe3+could be reduced into Fe2+, which I don’t think is the case.
Table 1, I don’t see this table is useful for the readers. It would be better to provide a summary description of what conditions researchers have studied and what had turned out working great.
Section “6. The genetic and genomic basis of FD and FT tolerance” Is there any mutant or over-expression analysis to highlight the direct roles these genes play in FD and FT tolerance, instead of only gene expression changes?
Figure 1, do the up arrows next to gene names indicate up-regulated expression under FT or FD condition? If so, please state that explicitly in the figure caption.
Line 513 to 527, the information provided in this paragraph is too general. For example, listing the differentially expressed miRNAs is not informative unless discussing their targets in the contexts of FT and FD.
Line 547 to 548, “DNA methylation is regulated by gene expression pattern by using methylation-sensitive amplified polymorphism (MSAP) techniques in barley” is not interpretable.
Line 554 to 555, I don’t think with the statement “methylation of DNA plays a crucial role in Fe-deprived and –sufficient conditions” is justified solely based on changes in pattern of methylation in response to FD. Unless a direct causal relationship is demonstrated, one can only conclude methylation of DNA is responsive to Fe conditions. Same logic applies to gene expression changes in response to different treatments.
Line 592, what is “IDs”?
Author Response
Reviewer 2
Comments and Suggestions for Authors
1. Graphical abstract, the current version contains way too much text, and is difficult to grasp the main points. Suggest reducing the level of details in the graph, such as listed examples of “other nutrients”, which could be removed from the graph. It would be nice to replace the current black and white rice plant photo with some color photos of the symptoms showing leaf bronzing and interveinal chlorosis. This will provide a more direct impression of the severity of FTS and FDS, respectively. The text “INTEGRATED APPROACH” is placed inside the photo, making it disconnected from the accompanying information to the right. Please relocate the text to make them better connected. Also, the orientation of some text is awkward, and please fix them in a way to make them easy to read.
Response: We have now corrected the graphical abstract as suggested to make it easier to read and understand the details on genotyping and phenotyping. We have provided the integrated approach in a horizontal way for easy understanding of breeding strategies for improving the tolerance of FTS and FDS conditions.
2. Throughout the manuscript, “Fe+2” and “Fe+3” should be changed into “Fe2+” and “Fe3+”.
Response: We have now changed “Fe+2” and “Fe+3” to Fe2+ and Fe3+ throughout the manuscript.
3. The Introduction section contains many contents that are redundant with the later sections in the manuscript. The introduction section should provide the necessary background knowledge on how significant FD and FT issues are on rice production, as well as a logical outline of contents of the manuscript. For example, many detailed description of FD and FT symptoms in the “1.
Response: We have revised the MS and detailed descriptions of FD and FT have been moved to the respective paragraphs in the MS.
4. The manuscript can be greatly condensed by reducing the aforementioned repetitiveness, and removing many uninformative statements, which include 1) writing down long lists of factors or gene names may be involved in responses to FT or FD without actually providing in depth discussion on the roles they play; 2) listing trivial numbers from other people’s study without giving a comprehensive summarization; 3) very general statements that is not directly related to FT and FD study, such as on line 540 to 542 “Among them, few proteins are validated for their biological functions, and many of them were unrevealed, and those may not be regulated at post-translation levels”.
Response: Thanks for your suggestions. We have revised the entire manuscript and also removed the repetitiveness in the MS (lines 64-65, 90-91, 186-189, 200-203, and 579-582).
5. The manuscript needs to highlight the take-home messages the authors want to deliver to their readers. The current version clearly lacks that, and may leave the audience feel confused or lost after reading through it.
Response: The aim of this manuscript preparation is to provide updated information on breeding for improving rice varieties with FD and FT. We have highlighted the details on the availability of QTLs and genes for application in breeding and also a basic understanding of the mechanisms involved in FD and FT. We have clearly mentioned the take-home message at the end of the introduction.
6. Line 46, suggest change “adding” into “additional”.
Response: As per your suggestion, we have now changed “adding” to “additional”.
7. Lines 63 to 65, the sentence “Affected by soil pH, reduced mobility of nutrients within the soil, and reduced soil-microbial interactions, overall poor soil health conditions affect the growth and survival of rice plants” is too general and not very informative. It could be removed.
Response: As per your suggestion, we have deleted the above-mentioned lines.
8. Line 65, suggest change into “About 30% and 18% of the global soil is Fe deficient and Fe toxic, respectively”
Response: As per your suggestion, we have now changed this.
9. Line 68, what is GY? I guess it means grain yield, but it is not defined in the entire manuscript.
Response: Thanks for your suggestion. We have now defined this in line 68.
10. Lines 88 to 93, it is confusing that the authors suddenly shifted the topic from FD to FT, while the prior portion of the paragraph has dedicated to FD. Why not move the FT contents to the following paragraph, where FT is discussed in greater details. Also, the sentence “Excess soluble Fe causes reduced cation exchange capacity, poor drainage, high sulfide content, and high organic matter content, leading to increased Fe availability, absorption, and uptake from the soil” is not interpretable. It is unclear from the writing whether high Fe content is the cause or consequence.
Response: We have now revised lines 83 to 111 in the MS as suggested to move the related information on FD and FT.
11. Line 101, please elaborate on “antagonistic effects on other essential nutrient uptake”.
Response: We have now elaborated on the antagonistic effects of other nutritional elements in the revised MS in lines 218 to 222.
12. Line 130, missing “the” in front of “Earth”. On the same line, please change “Fe oxides” into “ferric oxides”.
Response: The “the” is not needed in front of “Earth”. We have now made the other change as per your suggestion in the respective place.
13. Line 132, I cannot understand “0.2% to 55% mg kg-1”. Does this mean 0.002 to 0.55 mg per kg? Same issue applies to the following sentence. Please also be consistent with unit for iron content in soil. On Line 146, the unit used becomes mg per L.
Response: Thank for your suggestions. We have now corrected this as 0.2% to 55% (20,000 to 550,000 mg kg-1) and 0.4% to 27.3% (40,000 to 273,000 mg kg-1) in the MS.
14. Lines 149 to 150, the sentence “The higher concentration of Fe in the soil also decreases the availability and absorption of other nutrient elements such as phosphorus and potassium” seems mis-placed. It would read more fluent if following the sentence ending in “causing FT to plants” on line 147.
Response: We have now corrected lines 155-156.
15. Lines 169 to 172, the sentence “Being a major cofactor for several enzymes, Fe helps in the regulation of protein stability and also facilitates chemical reactions such as hydration/dehydration, hydroxylation, redox-dependent catalysis, and photo-redox catalysis, which are involved in the detoxification of excess Fe” implies Fe facilitates chemical reactions that detoxify excessive Fe, which does not sound intuitive. Providing an example of such reaction would help readers to understand the process. In addition, this sentence is highly redundant with the sentence on lines 176 to 177, which says “Fe acts as a cofactor for several enzymes, helps regulate protein stability, and also facilitates the several chemical reactions that are involved in detoxification”. Please delete either one.
Response: We have now deleted lines 176 to 177. Yes, we agree with that regarding generalized functions for Fe and related information has been added in the MS.
16. The sentence from lines 177 to 182 “Both deficiency and excess of Fe are detrimental to plants, but excess Fe is more damaging as it causes irreversible damage since it triggers the production of large amounts of reactive oxygen species and free radicals, which affect membrane lipids and proteins and oxidize chlorophyll and subsequently decrease photosynthesis and grain yield [80–82]. Because excess Fe is detrimental to plants, its mitigation through various strategies is necessary.” Should be moved to the end of this section, following the description of FD and FT symptoms and mechanisms.
Response: We have now shifted lines 177-182 to the end of the section. In the revised MS, the lines are 236-240.
17. Physiological basis of tolerance of FT and FD: ”How are FT and FD defined in rice cultivars?
Response: FD occurs in plants at an Fe concentration of less than 50 ppm of Fe, whereas more than 75 to 250 ppm concentration of Fe in plants is considered to represent FT.
18. Lines 215 to 216, “Increased protons lead to lower pH in the rhizosphere, thereby solubilizing Fe+3 to Fe+2” implies by lowering pH alone, Fe3+ could be reduced into Fe2+, which I don’t think is the case.
Response: We have now corrected the sentence as “Increased protons are acidifying the rhizosphere” instead of “Increased protons lead to lower pH in the rhizosphere”.
19. Table 1, I don’t see this table is useful for the readers. It would be better to provide a summary description of what conditions researchers have studied and what had turned out working great.
Response: Thanks for your suggestions, but we have given comprehensive information on FTS and FDS screening methodologies and Fe concentration could provide a valuable resource for researchers involved in screening and improving rice genotypes for FT and FD conditions.
20. Section “6. The genetic and genomic basis of FD and FT tolerance” Is there any mutant or over-expression analysis to highlight the direct roles these genes play in FD and FT tolerance, instead of only gene expression changes?
Response: We have now slightly modified Fig. 2. We elaborated on up- and downregulated gene expression patterns and the relationship to FT and FD was indicated in Fig. 2.
Yes, we have found that some of the genes and transcription factors are overexpressed on shoot and root analysis such as OsYSL15, OsNAS1, OsNAAT1, OsDPR, OsDMAS1, OsWRKY55, IDEF1, OsIRT1, and OsRab6a, which involves an increase in DMA biosynthesis, metal homeostasis, and secretion of mugineic acids. These genes were involved in uptake and transport of Fe from roots to shoots and grains (Kobayashi et al., 2007; Inoue et al., 2009; Zhang et al., 2012; Viana et al., 2017; Kar and Panda, 2018).
21. Figure 1, do the up arrows next to gene names indicate up-regulated expression under FT or FD conditions? If so, please state that explicitly in the figure caption.
Response: Thanks for your suggestion. We have now updated Fig. 2 and also we have changed the figure caption from “Diagram of Fe availability and transporting mechanism of FD- and FT-related genes in rice” to “Diagram of FT and FD tolerance genes, showing the changes in transcriptional levels (up-regulation or down-regulation indicated by arrows) through microarray and transcriptomics studies”.
22. Lines 513 to 527, the information provided in this paragraph is too general. For example, listing the differentially expressed miRNAs is not informative unless discussing their targets in the contexts of FT and FD.
Response: We have now added those miRNA functions in the MS.
Among these, few miRNAs are significantly involved in regulating metal transporters from roots. For instance, miR156 has a relationship with other metal transporters such as P, N, S, and Mn (Paul et al., 2015), miR171 for N and Zn (Paul et al., 2015), and miR162 for tolerance of cadmium (Mendoza-Soto et al., 2012).
23. Lines 547 to 548, “DNA methylation is regulated by gene expression pattern by using methylation-sensitive amplified polymorphism (MSAP) techniques in barley” is not interpretable.
Response: Now, we have corrected the sentence “In response to FD, one available report indicates that epigenetic mechanism of DNA methylation pattern of methylation-sensitive amplified polymorphism (MSAP) technique have been followed in barley (Hordeum vulgare L.)” to “In response to FD, one available report indicates that DNA methylation is regulated by gene expression pattern by using methylation-sensitive amplified polymorphism (MSAP) techniques in barley (Hordeum vulgare L.) [234]”. (This is now in lines 562 to 564, which could change.)
24. Lines 554 to 555, I don’t think with the statement “methylation of DNA plays a crucial role in Fe-deprived and –sufficient conditions” is justified solely based on changes in pattern of methylation in response to FD. Unless a direct causal relationship is demonstrated, one can only conclude methylation of DNA is responsive to Fe conditions. Same logic applies to gene expression changes in response to different treatments.
Response: We have now corrected the previous statement “Therefore, to understand the epigenetic mechanisms of histone modifications, DNA methylation pattern and chromatic structures may provide a better understanding of the molecular and physiological mechanism of imperative genes that are involved in Fe uptake and transportation, which may lead to adaptation under FD and FT conditions” to “Therefore, methylation of DNA plays a crucial role in Fe-deprived and -sufficient conditions, which helps in understanding the physiological and molecular imperative genes that are involved in Fe uptake, distribution, and translocation, probably by activating or silencing genes or transporters, which may lead to adaptation under FD and FT conditions”.
25. Line 592, what is “IDs”?
Response: IDs = Iron disorders, which has now been replaced with FDS and FTS conditions.
Reviewer 3 Report
This article contains many interesting informations regarding the basis of Fe toxicity and tolerance. The authors tried to focus some recent studies emphasizing the molecular bases and genetic studies. However, this article is the lack of general information. To make it more understandable to the readers I suggest the authors including some sections regarding the physiological role of Fe and their uptake translocation. Also, compare tolerant and susceptible genotypes - how they responded to Fe, etc...
How plant respond to Fe in different soil condition (pH, moisture etc.) should be considered.
Language quality should be improved.
Author Response
Reviewer 3
Comments and Suggestions for Authors
This article contains many interesting informations regarding the basis of Fe toxicity and tolerance. The authors tried to focus some recent studies emphasizing the molecular bases and genetic studies. However, this article is the lack of general information. To make it more understandable to the readers I suggest the authors including some sections regarding the physiological role of Fe and their uptake translocation. Also, compare tolerant and susceptible genotypes - how they responded to Fe, etc...
Response: We have clearly indicated the physiological role of Fe and its mechanism for uptake and the regulating mechanism in the section on the physiological basis of tolerance of FT and FD. The translocation of Fe from roots to shoots and genotype performance were given detailed information in Table 2 and in the section on “phenotypic screening and breeding for FDS and FTS”.
How plant respond to Fe in different soil condition (pH, moisture etc.) should be considered.
Response: Thank you for your suggestions. We have mentioned this in the MS in the section on Fe in the soil and we have explained the forms of availability of Fe uptake to the rice plant.
Language quality should be improved.
Response: The manuscript has been thoroughly edited by a professional English editor and the language has been more polished now.
Round 2
Reviewer 2 Report
Line 106, there is an additional “( “ that should be removed.
The authors’ response letter indicated line 236 to 240 should read “Both deficiency and excess of Fe are detrimental to plants, but excess Fe is more damaging as it causes irreversible damage since it triggers the production of large amounts of reactive oxygen species and free radicals, which affect membrane lipids and proteins and oxidize chlorophyll and subsequently decrease photosynthesis and grain yield [78–80]. Because excess Fe is detrimental to plants, its mitigation through various strategies is necessary.” However, it is not the case. And the same exact quoted content appeared twice elsewhere, on line 150 to 154, and on line 190 to 195.
Line 160 to 166, this section is hard to follow, abruptly switching from IVC in iron deficiency to Fe toxicity, and contains redundant information just appeared on line 151 to 153, and later again on line 167 to 169.
Author Response
Comments and Suggestions for Authors
Reviewer 1
1. Line 106, there is an additional “(“ that should be removed.)
Response: Thanks for your suggestions. We have now removed “(“ in the revised manuscript at line number 141.
2. The authors’ response letter indicated line 236 to 240 should read “Both deficiency and excess of Fe are detrimental to plants, but excess Fe is more damaging as it causes irreversible damage since it triggers the production of large amounts of reactive oxygen species and free radicals, which affect membrane lipids and proteins and oxidize chlorophyll and subsequently decrease photosynthesis and grain yield [78–80]. Because excess Fe is detrimental to plants, its mitigation through various strategies is necessary.” However, it is not the case. And the same exact quoted content appeared twice elsewhere, on line 150 to 154, and on line 190 to 195.
Response: We have now deleted the same information quoted in the line number 192 to 196 and also we have placed it as per your earlier suggestions at end of the “Role of Fe in plants” section at line number 239-243. Now, we have modified the sentence in the same line.
3. Line 160 to 166, this section is hard to follow, abruptly switching from IVC in iron deficiency to Fe toxicity, and contains redundant information just appeared on line 151 to 153, and later again on line 167 to 169.
Response: Thanks for your suggestions. We have now revised the line number from 197 to 201 for Fe deficiency, whereas in FT also, we have now corrected in the line number 211-212 in the revised MS. In the revised MS, we have now excluded redundant information from 219-221 line numbers.

Reviewer 3 Report
This version is improved.
Author Response
Reviewer 3
1. A well-executed review one the importance of iron deficiency and -tolerance in rice.
2. A table of abbreviations with their explanations right after the abstract would make the manuscript more clear and readable.
Response: Thanks for your suggestions. Now, we have expanded the abbreviations and changed the orientation of table in the revised manuscript.
